# UVR8-mediated inhibition of shade avoidance involves HFR1 stabilization in Arabidopsis

**Eleni Tavridou**[ID][1,2], **Emanuel Schmid-Siegert**[ID][3], **Christian Fankhauser**[4], **Roman Ulm**[ID][1,2]*

**1** Department of Botany and Plant Biology, Section of Biology, Faculty of Science, University of Geneva, CH, Geneva, Switzerland, **2** Institute of Genetics and Genomics of Geneva (iGE3), University of Geneva, Geneva, Switzerland, **3** SIB-Swiss Institute of Bioinformatics, University of Lausanne, CH, Lausanne, Switzerland, **4** Center for Integrative Genomics, Faculty of Biology and Medicine, University of Lausanne, CH, Lausanne, Switzerland

* roman.ulm@unige.ch

**Data Availability Statement:** RNA-Seq data reported in this article have been deposited in NCBI's Gene Expression Omnibus and are accessible through GEO Series accession number GSE146125.

## Abstract

Sun-loving plants perceive the proximity of potential light-competing neighboring plants as a reduction in the red:far-red ratio (R:FR), which elicits a suite of responses called the "shade avoidance syndrome" (SAS). Changes in R:FR are primarily perceived by phytochrome B (phyB), whereas UV-B perceived by UV RESISTANCE LOCUS 8 (UVR8) elicits opposing responses to provide a counterbalance to SAS, including reduced shade-induced hypocotyl and petiole elongation. Here we show at the genome-wide level that UVR8 broadly suppresses shade-induced gene expression. A subset of this gene regulation is dependent on the UVR8-stabilized atypical bHLH transcription regulator LONG HYPOCOTYL IN FAR-RED 1 (HFR1), which functions in part redundantly with PHYTOCHROME INTERACTING FACTOR 3-LIKE 1 (PIL1). In parallel, UVR8 signaling decreases protein levels of the key positive regulators of SAS, namely the bHLH transcription factors PHYTOCHROME INTERACTING FACTOR 4 (PIF4) and PIF5, in a COP1-dependent but HFR1-independent manner. We propose that UV-B antagonizes SAS via two mechanisms: degradation of PIF4 and PIF5, and HFR1- and PIL1-mediated inhibition of PIF4 and PIF5 function. This work highlights the importance of typical and atypical bHLH transcription regulators for the integration of light signals from different photoreceptors and provides further mechanistic insight into the crosstalk of UVR8 signaling and SAS.

## Author summary

Sunlight provides the energy that powers photosynthesis and is thus of the utmost importance for plant growth and development. Plants grow in constantly changing environments, including highly variable light conditions. They have evolved specific light sensors (photoreceptors) that allow optimization of photosynthesis and survival, and adjusting growth and development to the prevailing environment. An intriguing example is the perception of competitors due to changes in the light transmitted and reflected by their leaves. As red light, but not far-red, is strongly absorbed by photosynthetic pigments, the red:far-red ratio (R:FR) is strongly reduced in the presence of competitors which is sensed

**Funding:** This work was supported by the University of Geneva and the Swiss National Science Foundation (grants no. 31003A_175774 to RU, and CRSII3_154438 to RU and CF). The funders had no role in study design, data collection and analysis, decision to publish, or preparation of the manuscript.

**Competing interests:** The authors have declared that no competing interests exist.

by the phytochrome red/far-red photoreceptors. Detection of competitors results in stem elongation in shade-intolerant plants to overtop the neighbors and reach full sunlight (shade avoidance syndrome). UV-B sensed by the UVR8 photoreceptor inhibits shade avoidance by reducing the abundance and activity of specific transcriptional regulators PIF4 and PIF5. We have discovered an important role for an atypical transcriptional regulator HFR1 that is stabilized under UV-B, strongly contributing to the UVR8 photoreceptor-mediated repression of shade responses by countering PIF4 and PIF5 activities. This contributes significantly to a deeper mechanistic understanding of how plants integrate information from a complex light environment.

## Introduction

Plants have evolved sophisticated sensory systems to respond and adapt appropriately to the environment. Light, in addition to being an energy source, is an important source of information about the environment for plants. Specific photoreceptors allow the detection of changes in light quality, duration, direction, and intensity, which is crucial to optimize photosynthesis, growth, and development, as well as to cope with light stress [1,2]. For example, plants growing in close proximity to their neighbors are able to sense that light resources may become limiting and preemptively elicit a suite of responses to avoid future shading, collectively known as shade avoidance syndrome (SAS) [3,4]. SAS promotes petiole and stem elongation so that a plant can outcompete its neighbors for light. SAS can be triggered in high PAR (photosynthetically active radiation) conditions through alterations in spectral composition and due to foliage properties. Photosynthetic pigments strongly absorb red (R) and blue (B) light and preferentially reflect or transmit far-red (FR) light. Thus, plants perceive a reduction in R:FR due to the scattering of FR from non-shading neighbors and initiate SAS [3,5].

Changes in R:FR are perceived by phytochrome B (phyB), which interconverts between an inactive (PrB) and an active (PfrB) state. Under high R:FR, the phyB photoequilibrium is shifted towards the active PfrB conformer that interacts with bHLH transcription factors of the Phytochrome Interacting Factor (PIF) family, including PIF4 and PIF5 that play a major role in shade responses [4,6,7]. Upon interaction with active PfrB, PIF4 and PIF5 are phosphorylated and targeted for proteasomal degradation, hence reducing the expression of shade response-associated genes [8,9]. However, under low R:FR, phyB is primarily in the inactive PrB state and unable to interact with and repress PIFs, thus allowing PIF-mediated activation of shade-response genes [8]. PIFs' main targets are other transcription regulators mainly involved in auxin signaling and cell elongation [10] as well as additional bHLH transcription regulators including PHYTOCHROME INTERACTING FACTOR 3-LIKE 1 (PIL1) and LONG HYPOCOTYL IN FAR-RED1 (HFR1). HFR1 provides a negative feedback loop by interacting with PIF4 and PIF5 and forming non-functional heterodimers that can no longer bind to DNA, thus preventing an exaggerated shade response [11].

Ultraviolet-B (UV-B) is perceived by the UV RESISTANCE LOCUS 8 (UVR8) photoreceptor [12]. Upon UV-B exposure, UVR8 monomerises and interacts with CONSTITUTIVELY PHOTOMORPHOGENIC 1 (COP1) to trigger downstream signaling [12–14]. COP1 is an E3 ubiquitin ligase that targets for proteasomal degradation transcriptional regulators that generally promote light responses and photomorphogenesis [15–17]. Interaction of activated UVR8 with COP1 leads to COP1 inactivation and thus accumulation of COP1 target proteins, such as the bZIP transcription factor ELONGATED HYPOCOTYL 5 (HY5) [13,18,19]. Stabilized HY5 is crucial for the regulation of numerous UV-B-induced genes, including those that

function in UV-B acclimation and tolerance [14,20–26]. UV-B photomorphogenesis involves the inhibition of hypocotyl elongation and in general elicits opposite responses to shade signals [13,27–31]. For example, UV-B was shown to antagonize auxin signaling and suppress shade-induced hypocotyl and petiole elongation, as well as shade marker gene induction [28,32–35]. Molecularly, UVR8 activity was associated with the degradation of PIF4 and PIF5 [27,28,32,36], the key inducers of SAS [3,8,37]. It was recently postulated that COP1 directly interacts with PIF5, causing PIF5 stabilization by an unknown mechanism [32]. Active UVR8 disrupts this stabilization through interaction with COP1, triggering PIF5 instability under UV-B [32]. In parallel, UV-B-stabilized DELLA proteins may form non-DNA-binding hetero-dimers with PIFs and stabilized HY5 may compete with PIFs for the same promoters [28,38–41]. Notwithstanding the abovementioned studies, the molecular mechanisms by which UVR8 signaling antagonizes SAS remain poorly understood.

Here we show that UV-B-activated UVR8 leads to the stabilization of the COP1 substrate HFR1, partially accounting for the UV-B-mediated suppression of shade marker gene induction. Furthermore, our data suggest that HFR1 acts cooperatively with PIL1 in the UV-B-mediated suppression of SAS. Moreover, PIF4 and PIF5 binding to promoters of shade-induced genes is reduced under supplemental UV-B, in agreement with the activity of HFR1 as well as the degradation of PIF4 and PIF5 under these conditions.

## Results

### UVR8-dependent stabilization of HFR1 suppresses induction of shade marker genes

As HFR1 is a COP1 substrate and activated UVR8 inactivates COP1 [17,18,42–45], we hypothesized that HFR1 stability is regulated by UVR8 in response to UV-B, which may contribute to the antagonistic effect of UV-B on SAS. We thus tested the effect of UV-B and low R:FR on *HFR1* promoter-driven HFR1-HA protein levels in an *uvr8* null mutant background compared to in a wild-type background. Seven-day-old seedlings were irradiated at Zeitgeber time ZT3 with 3-h supplemental UV-B (+UVB), 3-h supplemental FR to create low R:FR (+FR), or a combination of the two treatments (+UVB+FR), and compared to seedlings maintained in white light as control (WL; -UVB-FR). Low R:FR treatment slightly increased HFR1-HA protein level, both in the absence and presence of UVR8, whereas HFR1 exhibited greater stabilization under UV-B which was UVR8 dependent (Fig 1A). Moreover, UV-B and low R:FR stabilized HFR1 in an additive manner (Fig 1A). HFR1 accumulation was associated with only a weak transcriptional activation of *HFR1* under UV-B (S1 Fig). These results indicate that active UVR8 signaling leads to post-translational HFR1 stabilization, likely through COP1 inhibition.

To test the contribution of HFR1 to UVR8-mediated suppression of low R:FR response, we analyzed shade-induced expression of shade marker genes *PIL1*, *XYLOGLUCAN ENDO-TRANSGLYCOSYLASE 7* (*XTR7*), *ARABIDOPSIS THALIANA HOMEOBOX PROTEIN 2* (*ATHB2*), and *INDOLE-3-ACETIC ACID INDUCIBLE 29* (*IAA29*) in *uvr8* and *hfr1* mutants in comparison to that in wild type. In each genotype, all four tested genes showed induced expression in response to low R:FR (+FR, shade), which was strongly suppressed by supplemental UV-B (+FR+UVB) in wild type but not in *uvr8* (Fig 1B–1E). Interestingly, UV-B suppression of shade-induced *PIL1* and *ATHB2* expression was also absent in *hfr1*, comparable to that in *uvr8* (Fig 1B and 1D). However, the HFR1-dependent UV-B effect was found to be gene specific, as the other tested shade marker genes *XTR7* and *IAA29* were repressed by UV-B in *hfr1* (Fig 1C and 1E). These data show that HFR1 is required to repress a subset of shade-induced genes in response to UV-B.

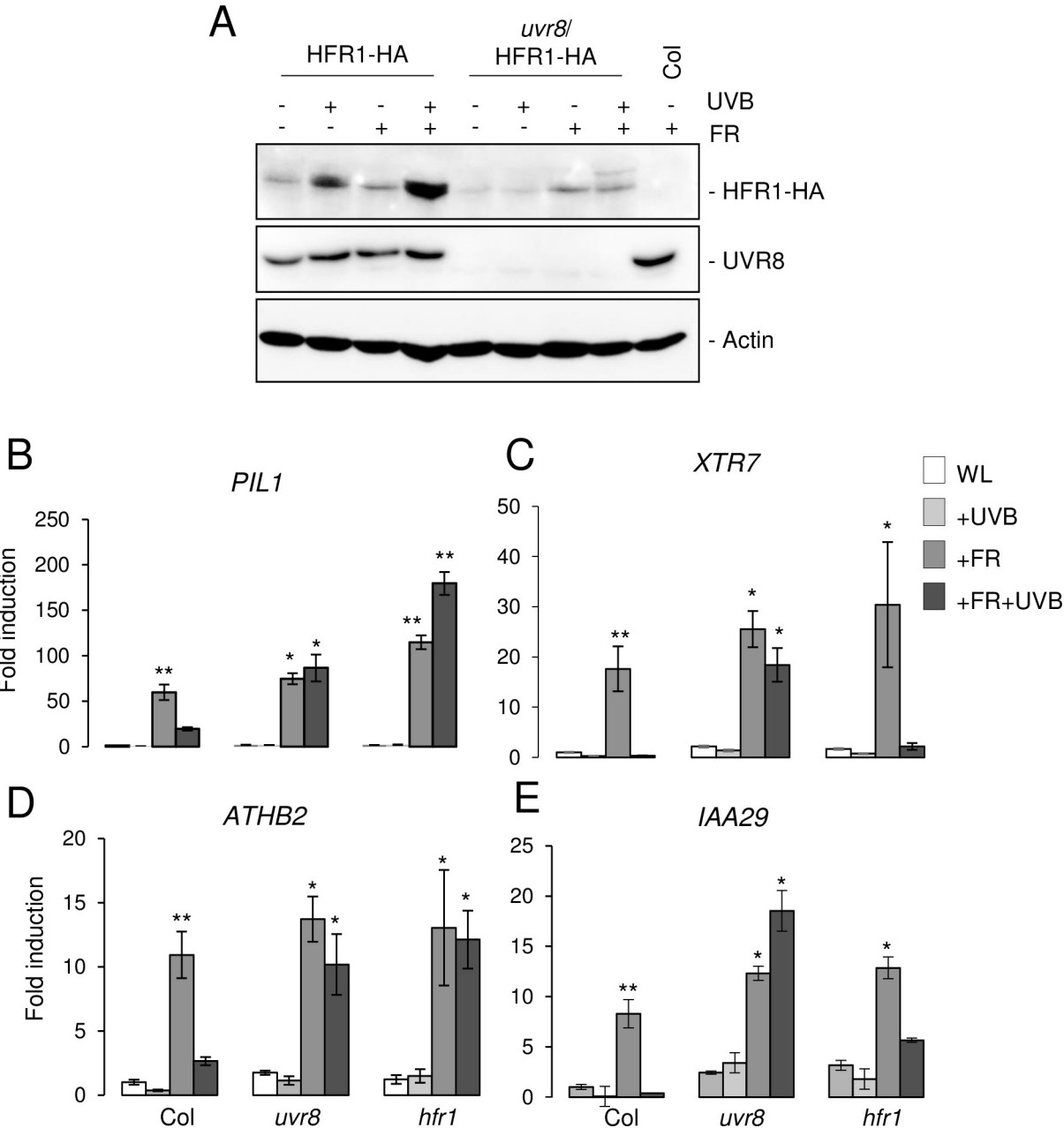

**Fig 1. UVR8-dependent stabilization of HFR1 suppresses activation of shade marker genes.** (A) Anti-HA immunoblot analysis of HFR1-HA levels in 7-day-old Col/Pro$_{HFR1}$:HFR1-3xHA (HFR1-HA) and *uvr8-6*/Pro$_{HFR1}$:HFR1-3xHA (*uvr8*/HFR1-HA) seedlings grown in long-day conditions under white light (WL) or white light supplemented at ZT3 with 3-h UV-B (+UVB), low R:FR (+FR), or a combination of both (+FR+UVB). Wild type (Col) is shown as negative control. The immunoblot was re-probed with anti-UVR8, as well as anti-actin as loading control. (B–E) RT-qPCR analysis of (B) *PIL1*, (C) *XTR7*, (D) *ATHB2*, and (E) *IAA29* expression in 7-day-old wild-type (Col), *uvr8-6*, and *hfr1-101* seedlings grown under white light and exposed to either UV-B (+UVB), low R:FR (+FR), or both (+FR+UVB) for 3 h or maintained under white light (WL). Error bars represent SEM of three biological replicates. Asterisks indicate a significant difference in transcript abundance compared to that under WL in each genotype (* p < 0.05; ** p < 0.01).

## UVR8 broadly suppresses shade gene induction

To investigate the extent of UVR8- and HFR1-dependent UV-B repression of shade-induced gene expression at the genome-wide level, we performed an RNA-Seq analysis comparing *uvr8*

and *hfr1* transcriptomes with that of wild type under WL, +FR, +UVB, and +FR+UVB conditions. Shade significantly induced 315, 330, and 365 genes in wild type, *uvr8*, and *hfr1*, respectively (fold change > 2 and p < 0.05, with FDR 5%), with an overlap of 204 robustly shade-induced genes (Fig 2A and 2B, S1A–S1C and S2 Tables). The shade induction of 77% of the shade-induced genes (+FR versus WL) was repressed by supplemental UV-B in wild type (i.e. no or less induction under +FR+UVB versus +FR) (Fig 2A). This repression by UV-B was UVR8 dependent, as a minimal proportion of shade-induced genes was UV-B suppressed in *uvr8* mutants (<2%) (Fig 2A). In *hfr1*, 58% of the shade-induced genes were repressed by UV-B treatment, which is significantly less than in wild type (Fig 2A). Indeed, among the 204 shade-induced genes common to all three genotypes, a cluster of 44 genes was found to be similarly regulated as *PIL1*: shade induced and UVB suppressed in wild type, but not in *uvr8* and *hfr1* (herein referred to as the *PIL1*-like cluster) (Fig 2C, S3 Table). In two independent *hfr1* mutant lines, reverse transcription quantitative PCR (RT-qPCR) analysis of two selected genes from the *PIL1*-like cluster, *SULFOTRANSFERASE 2A* (*ST2A*) and *FLOWERING PROMOT-ING FACTOR 1-LIKE PROTEIN 1* (*FLP1*), supported an essential role of HFR1 in the UV-B-repressive effect on their shade-induced expression (Fig 2D and 2E). Our data suggest that the antagonizing effect of UV-B on shade-responsive gene expression is broad at the genome-wide level, fully dependent on UVR8, and dependent on HFR1 for a subset of genes.

## HFR1 and PIL1 act redundantly in UV-B suppression of shade-induced gene expression

Similar to HFR1, PIL1 is a negative regulator of shade responses and targeted for proteasomal degradation by COP1 [46,47]. As *PIL1* is not UV-B repressed in *hfr1* mutants, we tested whether PIL1 plays a redundant role with HFR1, particularly in the regulation of the subset of genes that are UV-B repressed in an HFR1-independent manner, as exemplified by the *FAR-RED-ELONGATED HYPOCOTYL1-LIKE* (*FHL*) gene. In agreement with the RNA-Seq data, *FHL* expression was induced under +FR and this expression induction was repressed by UV-B under +FR+UVB in wild type as well as in *pil1* and *hfr1* single mutants, although to a reduced extent in the latter (Fig 3A). By contrast, however, *FHL* shade-induced expression was not reduced by UV-B in the *hfr1 pil1* double mutant (Fig 3A). This was observed to a much lesser extent for two further HFR1-independent marker genes, *YUCCA8* (*YUC8)* and *GH3.3* (S2A and S2B Fig). *PIL2*, a marker gene for which the UV-B-repressive effect on its shade-induced expression depends on HFR1, was typically repressed in the *pil1* single mutant but hyper-induced in the *hfr1 pil1* double mutant under +FR+UVB (Fig 3B). These results show that HFR1 and PIL1 play a redundant role in the UV-B repression of, at least, shade-induced *FHL* expression.

## UVR8 negatively regulates PIF4 and PIF5 levels and chromatin association

In order to understand how activated UVR8 may affect the key shade-response regulators PIF4 and PIF5, we tested the effect of UV-B on protein levels of PIF4-HA and PIF5-HA expressed under the constitutive CaMV 35S-promoter. In agreement with previous data [8,28], PIF4-HA and PIF5-HA accumulated under low R:FR (+FR) and UV-B severely decreased their protein abundances under +FR+UVB conditions (Fig 4A and 4B). UV-B-induced PIF4 and PIF5 degradation was shown to be UVR8 dependent, as their levels were not reduced in the absence of UVR8 (Fig 4A and 4B).

We used *pif4*/PIF4-HA and *pif4 bop2*/PIF4-HA lines to determine whether the BOP2 E3 ubiquitin ligase [48] was involved in UV-B-induced PIF4 degradation. However, although higher levels of PIF4-HA were detected in the absence of *BOP2* under +FR conditions as

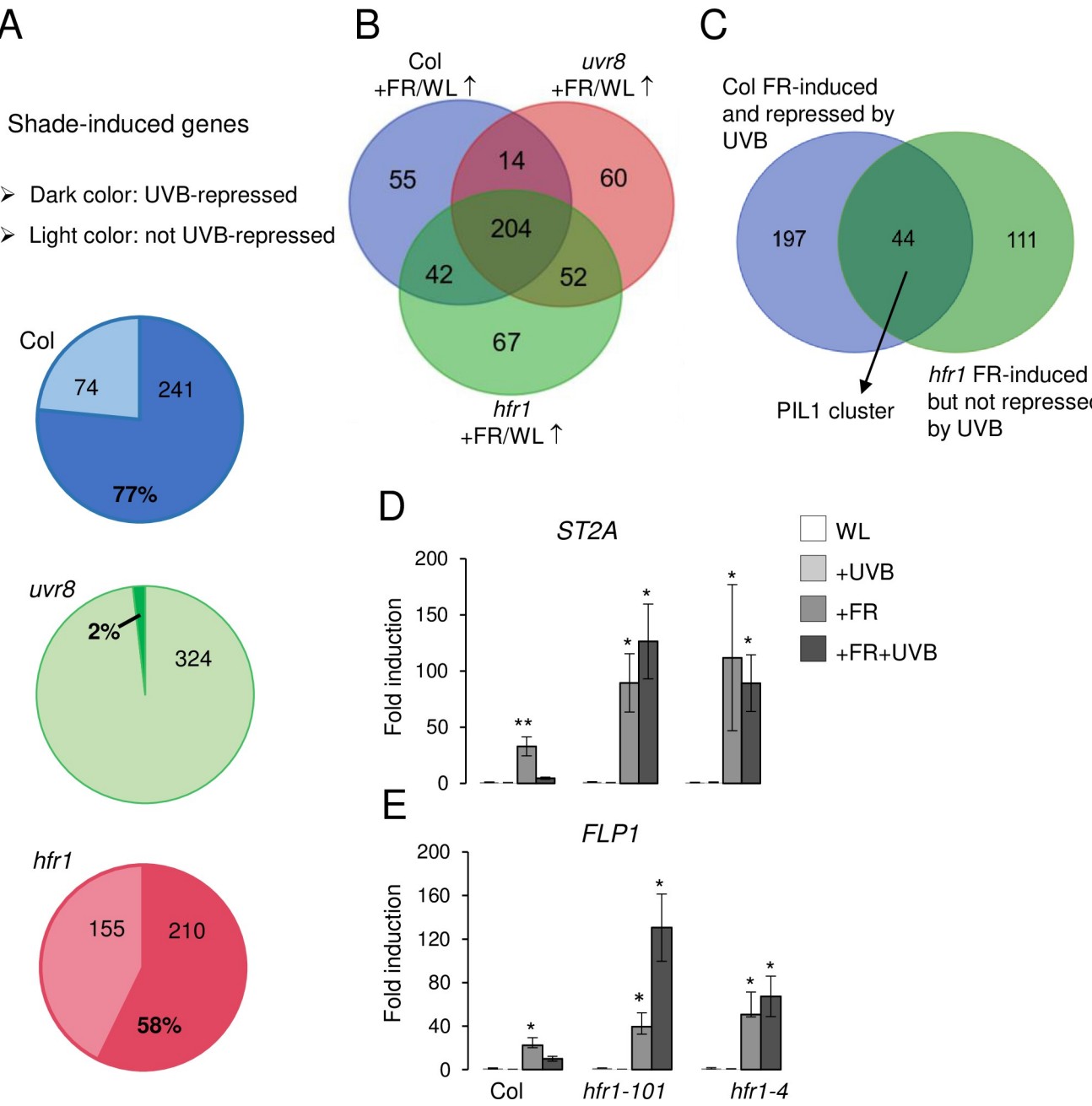

**Fig 2. UV-B-activated UVR8 broadly suppresses shade-responsive gene expression.** (A) Pie charts showing the percentage and number of shade-induced genes that are repressed by UV-B (FC > 2, adjusted pvalue < 0.05) in wild type (Col), *uvr8-6*, and *hfr1-101* seedlings. (B) Venn diagram representing the intersection between shade-induced genes in wild type (Col +FR/WL ↑), *uvr8-6* (*uvr8* +FR/WL↑), and *hfr1-101* (*hfr1* +FR/WL ↑) seedlings (FC > 2, adjusted pvalue < 0.05). (C) Venn diagram representing the intersection between shade-induced genes that are repressed by UV-B in a UVR8-dependent manner in wild type (Col) but not in *hfr1-101* (FC > 2, adjusted pvalue < 0.05). The intersection of 44 genes is referred to as the *PIL1* cluster. (D, E) RT-qPCR analysis of (D) *ST2A* and (E) *FLP1* expression in 7-day-old Col, *hfr1-101*, and *hfr1-4* seedlings grown under the indicated light conditions for 3 h. Error bars represent SEM of three biological replicates. Asterisks indicate a significant difference compared to that under WL in each genotype (* p < 0.05; ** p < 0.01).

expected, supplemental UV-B severely decreased PIF4-HA levels in both the *bop2* and wild-type backgrounds (Fig 4C). Moreover, PIF4-HA levels were reduced in *cop1-4* mutants compared to that in a wild-type background under +FR, but no UV-B effect could be detected in

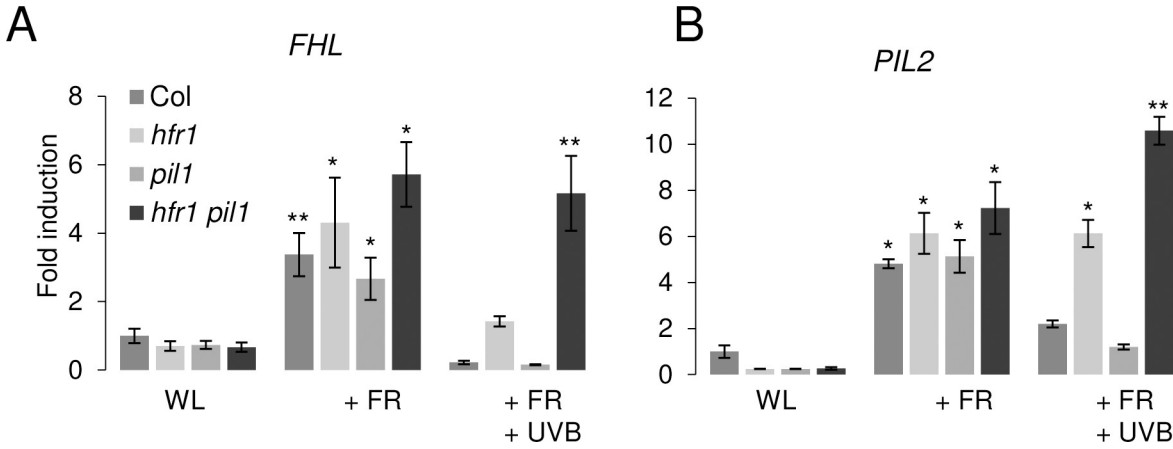

**Fig 3. PIL1 and HFR1 act redundantly in UVR8-mediated repression of shade-induced *FHL* and *PIL2* expression.** (A, B) RT-qPCR analysis of (A) *FHL* and (B) *PIL2* expression in 7-day-old seedlings of wild type (Col), *hfr1-101*, *pil1-6*, and *hfr1-101 pil1-6* grown under constant white light (WL) and transferred to the indicated experimental light conditions for 3 h. Error bars represent SEM obtained from three biological replicates. Asterisks indicate the statistical significance compared to WL in each genotype (* $p < 0.05$; ** $p < 0.01$).

*cop1-4* in agreement with the role of COP1 in UV-B signaling (Fig 4D) [14]. Furthermore, *PIF4* and *PIF5* expression levels were found to be downregulated by UV-B in a UVR8-dependent but HFR1-independent manner (S3A and S3B Fig). Our data suggest that activated UVR8 is responsible for the decrease in PIF4 and PIF5 levels under +FR+UVB conditions, which, at least in the case of PIF4, is independent of the E3 ubiquitin ligase BOP2.

In response to shade, PIF4 and PIF5 are known to directly bind to G-boxes in the promoters of shade-regulated genes. *PIL1* and *XTR7* are among those genes known to be direct targets of PIF4 and PIF5 [11,49,50]. Chromatin immunoprecipitation (ChIP) assays showed that UV-B, in a UVR8-dependent manner, abolished PIF4 and PIF5 association with regions of the *PIL1*, *XTR7* and *HFR1* promoters that contain G-boxes (Fig 5 and S4 Fig). Collectively, our data suggest that UVR8 signaling impairs PIF4 and PIF5 activities by eliciting their degradation.

## HFR1 is not involved in the UV-B-mediated downregulation of PIF4 and PIF5

We further tested whether HFR1 could affect the regulation of PIF4 and/or PIF5 protein levels, similar to what has been previously suggested for PIF1 and HFR1 [51]. However, in an *hfr1* background, PIF4-HA accumulation under +FR and degradation under +FR+UVB were comparable to that in the wild-type background (Fig 6A and 6B). In agreement, ChIP experiments showed that PIF4-HA chromatin association was reduced in both the *hfr1* and wild-type backgrounds under +FR+UVB compared to that under +FR (Fig 6C). Although the effect seems reduced in *hfr1* mutants, which would be in agreement with the known activity of HFR1 heterodimerizing with PIF4 and PIF5 [11], this has only been clearly observed in two out of three independent repetitions (Fig 6C, and S5E and S5F Fig). Notwithstanding this, these results illustrate that HFR1 is not required for the UVR8-mediated destabilization of PIF4 and PIF5. Thus, the absence of a UV-B effect on the shade-induced expression of *PIL1* in *hfr1* implies that the transcriptional regulation of *PIL1* and likely *PIL1*-like cluster genes involves additional molecular players.

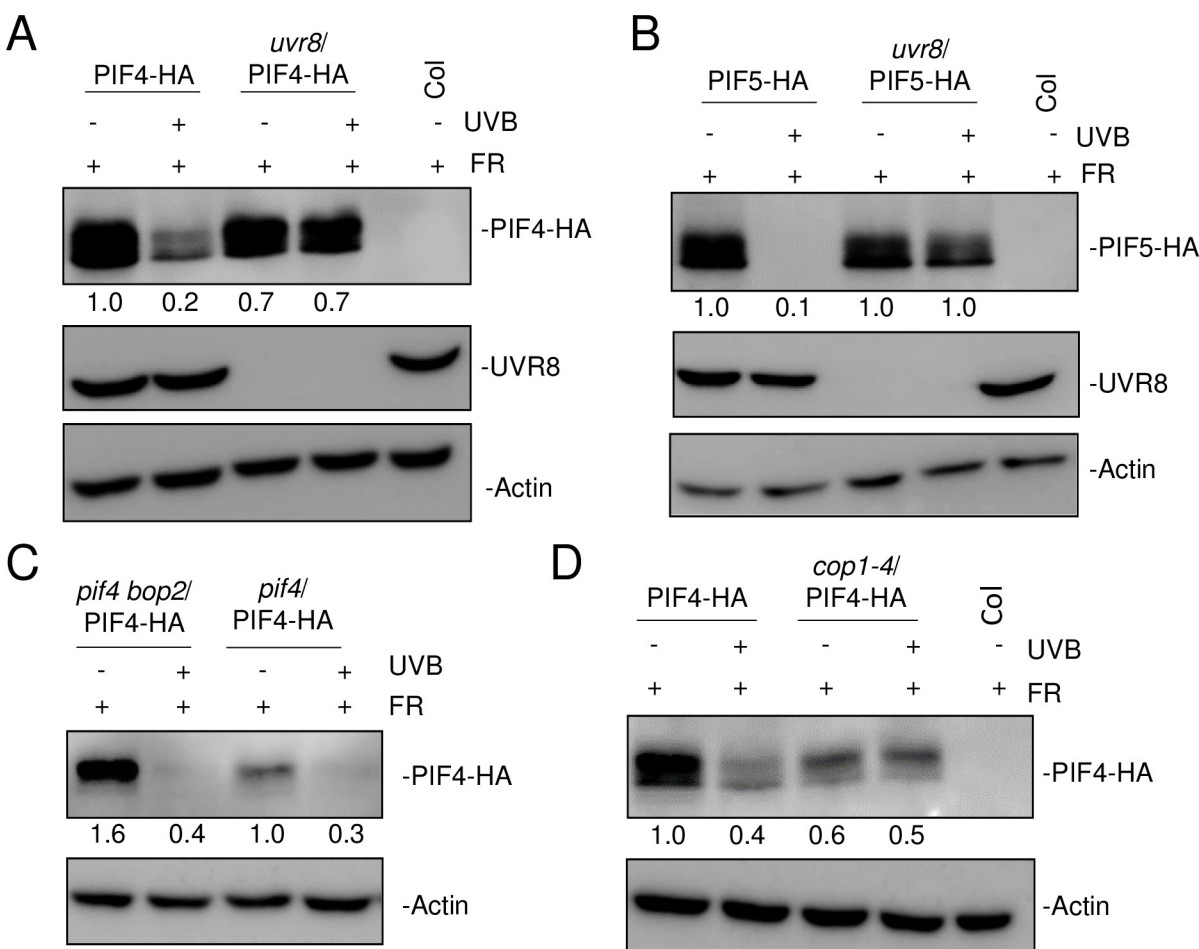

**Fig 4. UVR8 induces PIF4 and PIF5 degradation in response to UV-B, independently of BOP2.** Anti-HA immunoblot analysis of HA-tagged PIF4 and PIF5 proteins in 7-day-old seedlings grown in long-day conditions under white light and exposed to low R:FR with (+) or without (-) 3-h supplemental narrowband UV-B at ZT3. Wild type (Col) is shown as negative control. (A) PIF4-HA protein levels were analyzed in Col/Pro_PIF4:PIF4-3xHA (PIF4-HA) and uvr8-6/Pro_PIF4:PIF4-3xHA (uvr8/PIF4-HA). (B) PIF5-HA protein levels were analyzed in Col/Pro_PIF5:PIF5-3xHA (PIF5-HA) and uvr8-6/Pro_PIF5:PIF5-3xHA (uvr8/PIF5-HA). (C) PIF4-HA protein levels were analyzed in pif4-101/Pro_PIF4:PIF4-3xHA (pif4/PIF4-HA) and pif4-101 bop2-2/Pro_PIF4:PIF4-3xHA (pif4 bop2/PIF4-HA). (D) PIF4-HA protein levels were analyzed in Col/Pro_PIF4:PIF4-3xHA (PIF4-HA) and cop1-4/Pro_PIF4:PIF4-3xHA (cop1-4/PIF4-HA). Blots were re-probed with anti-UVR8 (A and B) as well as anti-actin as loading control (A–D). Numbers under lanes indicate relative band intensities.

## Discussion

UV-B provides an unequivocal signal for sunlight exposure that inhibits SAS, which is applicable such as upon emergence of plant tissues from a canopy. UV-B perceived by UVR8 broadly suppresses shade-induced gene expression, and inhibits shade-induced hypocotyl and petiole elongation [28,33, and this work]. This study further shows that UVR8-mediated inhibition of the E3 ubiquitin ligase COP1 results in the stabilization of its target protein HFR1, a negative regulator of PIF4 and PIF5 transcription factors. In addition to PIF4 and PIF5 degradation, this mechanism involving HFR1 strongly contributes to UV-B-mediated SAS suppression. Our observations thus identify HFR1 as a molecular effector of UVR8 photoreceptor signaling to antagonize plant shade avoidance.

PIF4 and PIF5 are key bHLH proteins for elongation growth, particularly in response to elevated temperature and shade [4,6,52,53]. The regulation of PIF4 and PIF5 includes their direct interaction with phytochrome and cryptochrome photoreceptors, as well as their light-

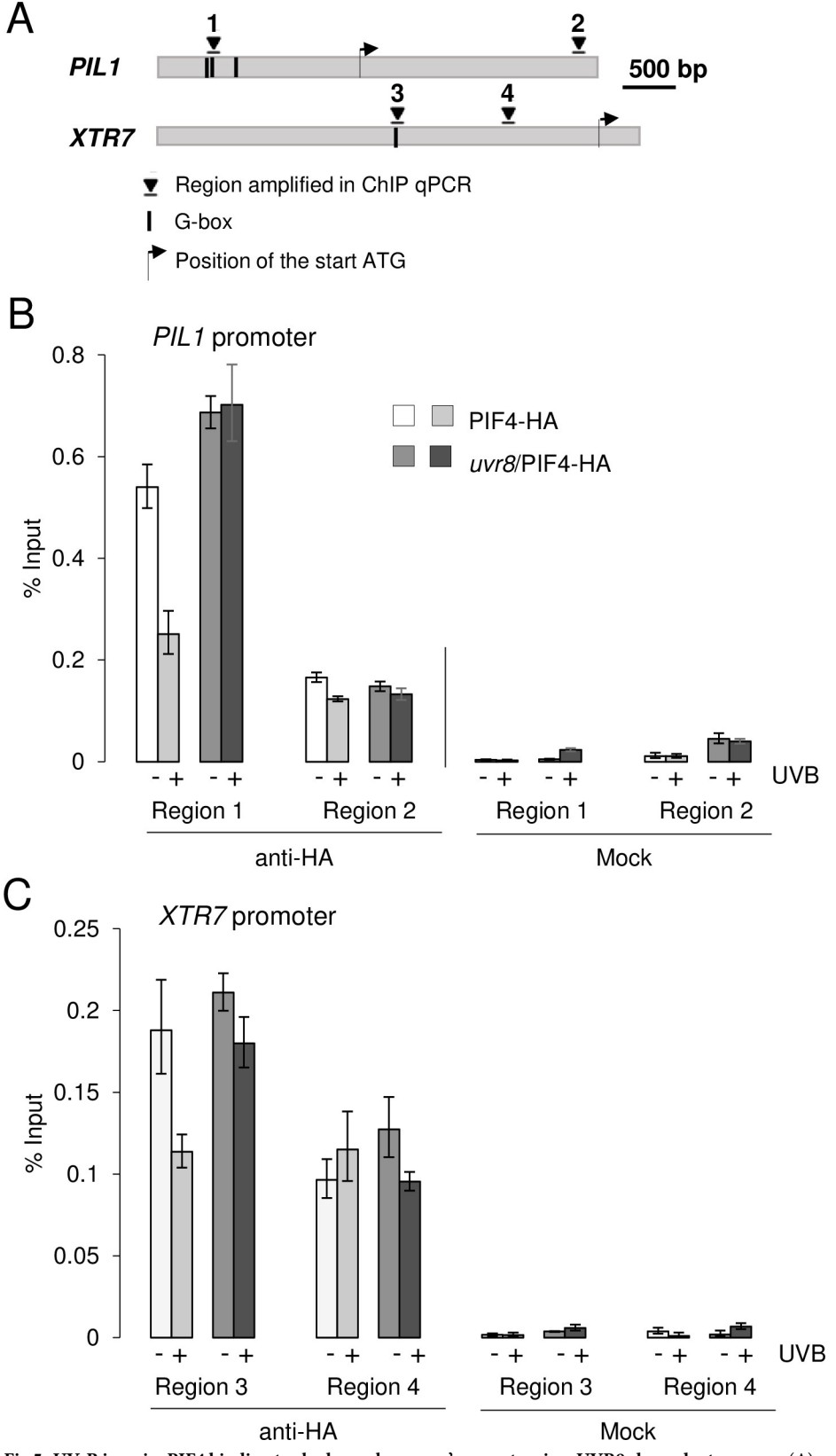

**Fig 5. UV-B impairs PIF4 binding to shade marker genes' promoters in a UVR8-dependent manner.** (A) Schematic representation of *PIL1* and *XTR7* with G-boxes and regions amplified in ChIP-qPCR indicated. (B, C)

PIF4-HA chromatin association in Col/Pro$_{PIF4}$:PIF4-3xHA (PIF4-HA) and *uvr8-6*/Pro$_{PIF4}$:PIF4-3xHA (*uvr8*/PIF4-HA). Ten-day-old seedlings were grown in long-day conditions under white light and exposed to low R:FR with (+) or without (-) 3-h supplemental narrowband UV-B at ZT3. ChIP-qPCR was performed for (B) *PIL1* and (C) *XTR7* promoters. ChIP data are presented as the percentage recovered from the total input DNA (% Input). Data shown are representative of three independent biological replicates (see S5A–S5D Fig). Error bars represent SD of three technical replicates. Immunoprecipitated DNA was quantified by qPCR using primers in the promoter regions containing G-boxes (region 1, Pro$_{PIL1\_-1417}$; region 3, Pro$_{XTR7\_-197}$) or control regions without G-boxes (region 2, Pro$_{PIL1\_+1816}$; region 4, Pro$_{XTR7\_-664}$).

dependent degradation [8,9,54–56]. Recently, the CUL3$^{BOP2}$ E3 ubiquitin ligase complex was shown to ubiquitinate PIF4, thus targeting it for proteasomal degradation [48]. PIF5, on the other hand, has recently been shown to be stabilized by the CUL4$^{COP1-SPA}$ E3 ubiquitin ligase complex in the dark and in shade, whereas CUL4$^{COP1-SPA}$ promotes PIF5 ubiquitination in response to red light [32,57]. Interestingly, activated UVR8 was found to disrupt COP1 stabilization of PIF5 in shade, which contributes to PIF5 ubiquitination and degradation under UV-B [27,28,32]. In contrast to phytochrome and cryptochrome mechanisms involving direct protein-protein interactions with PIF4 and PIF5, no direct interaction has been detected between these transcription factors and UVR8 [28,32]. Moreover, whereas UVR8 is known to interact with and inhibit COP1 in response to UV-B [13,17,18], which is linked to PIF5 destabilization [32], our data suggest that PIF4 degradation under UV-B is independent of BOP2, despite clear stabilization of PIF4-HA in the *bop2* background under our growth conditions that is in agreement with previously published data [48]. As PIF4 levels are also reduced in *cop1-4* mutants and no additional UV-B-mediated degradation is detectable in the absence of functional COP1, COP1-mediated stabilization of PIF4 may be similarly disrupted by activated UVR8, as previously suggested for PIF5 [32]. It is interesting to note that UV-B-mediated degradation of PIF4 is temperature dependent in that it was observed at 20˚C but not at the elevated temperature of 28˚C in previous work [36]. Notwithstanding this, the E3 ubiquitin ligase for PIF4 and PIF5 ubiquitination and proteasomal degradation under UV-B remains to be identified.

The UVR8 photoreceptor contains a functionally relevant C-terminal domain with a domain that mimics the COP1 interaction domains of COP1 substrates, including that of HFR1 [18,45,58]. Activation of UVR8 results in high-affinity cooperative binding of COP1 through its VP motif and its photosensory domain, which prevents binding of COP1 to HFR1 and thus results in HFR1 stabilization under UV-B [18]. Stabilized HFR1 inhibits the remaining non-degraded PIF4 and PIF5 pool under UV-B through heterodimer formation and inhibition of their binding to DNA [11], thus contributing to the antagonizing effect of UVR8 on both thermomorphogenesis as well as shade responses [36, and this work]. In addition to the two post-translational mechanisms of regulating PIF4 and PIF5 activities, namely degradation and stabilization of the negative regulator HFR1, *PIF4* and *PIF5* are also transcriptionally repressed by UVR8 and COP1 in response to UV-B [28,32,36] (S3 Fig). The resulting repression of PIF4 and PIF5 activities results in repression of auxin synthesis and responses causing reduced hypocotyl and petiole elongation, thus underlying how UV-B strongly antagonizing SAS and thermomorphogenesis [7,28,34,36].

*PIL1* is an early shade-responsive gene encoding the bHLH transcription factor PIL1, which is also targeted for ubiquitination and degradation by COP1 [11,47,59]. Although PIL1 is a well-established negative regulator of SAS, its mechanism of action remains poorly understood. PIL1 was found to interact with PIF5 and to form a negative feedback loop regulating the activity of its own promoter [46]. However, whether PIL1 competes with PIFs for binding to the same G-boxes or forms non-DNA-binding heterodimers, similar to HFR1, remains

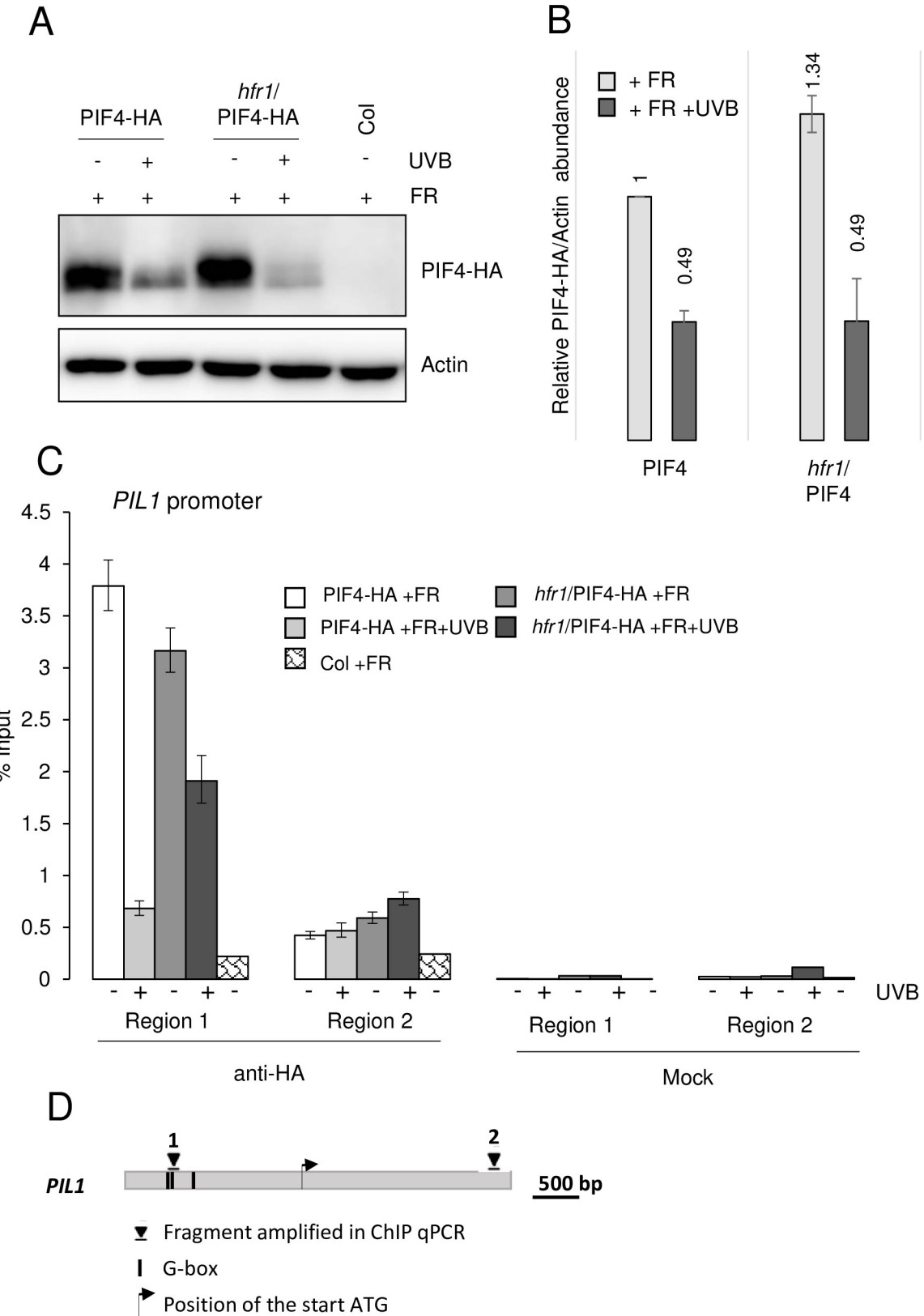

**Fig 6. UV-B-induced PIF4 degradation and reduced association with shade marker genes' promoters is independent of HFR1.** (A) Anti-HA immunoblot analysis of PIF4-HA in 7-day-old seedlings grown in long-day conditions under white light and exposed to 3-h

low R:FR (+FR) at ZT3 in the presence (+UVB) or absence of supplemental UV-B (-UVB). Wild type (Col) treated with low R:FR is shown as negative control. PIF4-HA protein levels were analyzed in Col/Pro$_{PIF4}$:PIF4-3xHA (PIF4-HA) and *hfr1-101*/Pro$_{PIF4}$:PIF4-3xHA (*hfr1*/PIF4-HA). Blot was re-probed with anti-actin as loading control. (B) Quantification of the immunoblot shown in (A). Error bars represent SD of three biological replicates. (C) Chromatin association of PIF4-HA in 10-day-old Col/Pro$_{PIF4}$:PIF4-3xHA (PIF4-HA) and *hfr1-101*/Pro$_{PIF4}$:PIF4-3xHA (*hfr1*/PIF4-HA) seedlings grown in long-day conditions under white light and exposed at ZT3 to 3-h low R:FR with (+) or without (-) supplemental UV-B. Col was included as negative control. ChIP-qPCR was performed for the *PIL1* promoter. ChIP of DNA associated with PIF4-HA is presented as the percentage recovered from the total input DNA (% Input). Data shown are representative of two independent biological replicates (see S5E and S5F Fig). Error bars represent SD of three technical replicates. Immunoprecipitated DNA was quantified by qPCR using primers in the promoter region containing a G-box (region 1, Pro$_{PIL1\_-1417}$) or control region without a G-box (region 2, Pro$_{PIL1\_+1816}$). (D) Schematic representation of *PIL1* with G-boxes and regions amplified in ChIP-qPCR indicated.

elusive [46,47]. Independent of the mechanistic aspects, we show that shade-induced *PIL1* expression is UV-B repressed in the absence of HFR1. Interestingly, we find that PIL1 acts in partial redundancy with HFR1 to repress shade marker genes in the presence of UV-B (Fig 3). Additional atypical bHLH transcription regulators could be involved in the UV-B suppression of shade responses, such as PHYTOCHROME RAPIDLY REGULATED 1 (PAR1) and PAR2, which are negative regulators of SAS with similar activities as HFR1 [60–62].

Collectively, we show that activated UVR8 broadly represses SAS, and that this is associated with negative regulation of PIF4 and PIF5 protein levels and activities, the latter likely involving HFR1 and PIL1 as inhibitors of PIF4 and PIF5. The involvement of additional PIF4 and/or PIF5 inhibitors in UVR8 repression of SAS remains to be investigated. In addition to the exposure of plants to full sunlight, SAS is also inhibited by sunflecks through phytochrome-, cryptochrome- and UVR8-mediated induction of the bZIP transcription factors HY5 and HYH [63,64]. The involvement of PIF4 and PIF5, as well as HFR1, under such conditions remains to be investigated and will contribute to our mechanistic understanding of SAS repression by a complex photoreceptor and signaling network.

## Materials and methods

### Plant material and generation of transgenic lines

The *uvr8-6* [13], *cop1-4* [65], *hfr1-101* (formerly *rsf1*) [66], *hfr1-4* [67], and *pil1-6* [46] mutants are in the Columbia (Col) accession. The *hfr1-101 pil1-6* double mutant was generated by genetic crossing followed by PCR genotyping of F2 plants (S4 Table).

*hfr1-101*/Pro$_{HFR1}$:HFR1-3xHA, *pif5-1*/Pro$_{PIF5}$:PIF5-3xHA, *pif4-101*/Pro$_{PIF4}$:PIF4-3xHA, and *pif4-101 bop2-2*/Pro$_{PIF4}$:PIF4-3xHA transgenic lines were described before [36, 48, 68]. *cop1-4*/Pro$_{PIF4}$:PIF4-3xHA, *uvr8-6*/Pro$_{PIF4}$:PIF4-3xHA, and *uvr8-6*/Pro$_{PIF5}$:PIF5-3xHA were generated by genetic crossing.

### Growth conditions and light treatments

Arabidopsis seeds were surface sterilized with sodium hypochlorite and grown on half-strength Murashige and Skoog basal salt medium (MS; Duchefa) containing 1% (wt/vol) phytagel (Sigma) and 1% (wt/vol) sucrose in either a 16-h/8-h light/dark cycle or continuous light, as indicated.

For gene expression analysis, seeds were stratified for at least 2 days at 4°C and were germinated at 22°C in a standard growth chamber (Percival Scientific) that contains LED lampbanks (CLF floralLED series, CLF Plant Climatics) under constant white light (R:FR = 4.7) of 60 µmol m$^{-2}$ s$^{-1}$ overall intensity measured using a LI-COR 250A light meter (LI-COR Biosciences). For low R:FR treatments, PAR remained constant with enrichment in FR provided by supplementary FR LEDs resulting in a R:FR ratio of 0.05. Supplementary UV-B was provided

by Philips TL20W/01RS narrowband tubes (0.06 mW cm$^{-2}$ s$^{-1}$; measured with a VLX-3W Ultraviolet Light Meter equipped with a CX-312 sensor, Vilber Lourmat). The UV-B range was modulated by the use of 3-mm transmission cut-off filters of the WG series (Schott Glaswerke).

For immunoblot analysis and ChIP assays, seeds were stratified for at least 2 days at 4˚C and were germinated at 22˚C in a standard growth chamber (MLR-350, Sanyo, Gunma, Japan) with 60 µmol m$^{-2}$ s$^{-1}$ white light and a 16-h/18-h light/dark cycle for 7 and 10 days, respectively. On day 7 or 10 at Zeitgeber Time 3 (ZT3), seedlings were transferred into a Percival growth chamber for a 3-h treatment under low R:FR (0.05) with or without supplemental UV-B (0.06 mW cm$^{-2}$ s$^{-1}$).

## Hypocotyl length measurements

Measurement of hypocotyl length was performed by ImageJ software (https://imagej.nih.gov/ij/). A minimum of 40 seedlings were measured per treatment and genotype, with at least two independent experimental repetitions. For each experiment, ANOVA type I statistical analysis was performed followed by Tukey HSD's post-hoc test. Letters were assigned to each statistically similar group with $p > 0.05$.

## Protein extraction and immunoblot analysis

Total proteins were extracted from Arabidopsis seedlings in extraction buffer (50mM Tris-HCl pH 7.6, 150mM NaCl, 5mM MgCl$_2$, 30% (v/v) glycerol, 10µM MG132, 10µM 3,4-dichloroisocoumarin, 1% (v/v) Sigma protease inhibitor cocktail, 0.1% (v/v) Igepal). Total protein was quantified by Bio-Rad Protein Assay (Bio-rad). Proteins were separated by SDS-PAGE and transferred to PVDF membranes according to the manufacturer's instructions (Bio-rad). Anti-HA (HA.11, Covance), anti-actin (Sigma-Aldrich), and anti-UVR8$^{(426–440)}$ [13] were used as primary antibodies, with horseradish peroxidase (HRP)-conjugated anti-mouse and anti-rabbit immunoglobulins used as secondary antibodies. Chemiluminescent signals were generated by using an Amersham ECL Select Western Blotting Detection Reagent kit (GE Healthcare, Little Chalfont, UK) and detected with an ImageQuant LAS 4000 mini CCD camera system (GE Healthcare).

## Reverse transcription quantitative PCR

Arabidopsis total RNA was isolated with Plant RNeasy kit (Qiagen) and treated with DNaseI according to the manufacturer's instructions. cDNA synthesis and quantitative PCR (qPCR) using PowerUpSYBR Green Master Mix reagents (Applied Biosystems) and a QuantStudio 5 real-time PCR system (Thermo Fisher Scientific) were performed as previously described [69]. Gene-specific primers used for *PIL1*, *XTR7*, *ATHB2*, *IAA29*, *ST2A*, *FLP1*, *FHL*, *PIL2*, *HFR1*, *GH3.3*, *YUC8*, *PIF4*, *PIF5*, and *PP2A* (reference gene) are listed in S4 Table. Expression values were calculated using the ΔΔCt method [70] and normalized to that in the wild type. Each reaction was performed in technical triplicates; data shown are from three independent biological repetitions.

## RNA-seq analysis

RNA quality control, library preparation using TruSeqUD Stranded mRNA (Illumina), and sequencing on an Illumina HiSeq 2500 system using 100-bp single-end reads protocol were performed at the iGE3 genomics platform of the University of Geneva. Quality control was performed with FastQC v.0.10.2. Reads were mapped to Arabidopsis TAIR-10 genome using

TopHat v2.0.13, resulting in an average alignment rate of approximately 92%. The BAM files were further processed with PicardTools v1.80 and counts obtained using HTSeq v.0.6.1. Counts were filtered for lowly expressed genes (to 20,926 genes) and normalized according to library size. Normalization and differential expression analysis was performed with the R/Bioconductor package edgeR v.3.4.2.

Differentially expressed genes were estimated using a GLM approach (General Linear Model), negative binomial distribution and a quasi-likelihood F-test. Pairwise comparisons (GLM, quasi-likelihood F-test) were performed on the filtered dataset with 2 factors defined (genotype and condition). Genes with a fold change > 2 and p < 0.05 (with FDR 5%) were considered differentially expressed. For further analysis, data sets were filtered for genes differentially expressed in response to UV-B in the wild type but not in *uvr8*. Venn diagrams were generated using a webtool (http://bioinformatics.psb.ugent.be/webtools/Venn/).

### Chromatin immunoprecipitation (ChIP) assays

Chromatin was immunoprecipitated with polyclonal ChIP-grade anti-HA antibodies (Abcam), as previously described [23,25]. ChIP-qPCR data were obtained using PowerUP SYBR Green Master Mix Kit and a QuantStudio 5 Real-Time PCR system (Applied Biosystems), with primers as listed in S4 Table. qPCR data were analyzed according to the percentage of input method [71]. Technical error bars represent standard deviation and were calculated according to the Applied Biosystems user manual.

## Supporting information

**S1 Fig. *HFR1* expression in response to UV-B.** (A) RT-qPCR analysis of *HFR1* expression in 4-day-old wild-type (Col) seedlings exposed to narrowband UV-B for 1 and 3 hours (1-h/3-h UVB) or not (WL). Error bars represent SEM of three independent biological replicates. Asterisks indicate a significant increase in transcript abundance compared to that under WL (*p < 0.05). (B) RT-qPCR analysis of *HFR1-3xHA* expression in 4-day-old seedlings of Col/Pro$_{HFR1}$:HFR1-3xHA (HFR1-HA) and *uvr8-6*/Pro$_{HFR1}$:HFR1-3xHA (*uvr8*/HFR1-HA) grown under white light (WL). Error bars represent SE of three independent biological replicates. (PDF)

**S2 Fig. PIL1 and HFR1 act redundantly in UVR8-mediated repression of shade-induced *YUC8* and *GH3.3* expression.** (A, B) RT-qPCR analysis of (A) *YUC8* and (B) *GH3.3* expression in 7-day-old wild-type (Col), *hfr1-101* (*hfr1*), *pil1-6* (*pil1*), and *hfr1-101 pil1-6* (*hfr1pil1*) seedlings grown under white light (WL) and exposed to 3-h low R:FR (+FR) or to 3-h low R: FR with supplemental UV-B (+FR +UVB), compared to seedlings maintained under WL as control. Error bars represent SEM of three independent biological replicates. Asterisks indicate a significant difference in transcript abundance compared to that under WL in each genotype (* p < 0.05; ** p < 0.01). (C) Hypocotyl length measurements of Col, *hfr1*, *pil1*, and *hfr1 pil1* seedlings grown in long-day conditions for 3 days before being transferred to WL, +UVB, +FR, and +FR+UVB for 4 days. Data represent mean length ± SE (n ≥ 40). (PDF)

**S3 Fig. HFR1 does not play a role in UVR8-dependent transcriptional repression of *PIF4* and *PIF5* under UV-B.** (A, B) RT-qPCR analysis of (A) *PIF4* and (B) *PIF5* expression in 7-day-old wild-type (Col), *uvr8-6*, and *hfr1-101* seedlings grown under white light (WL) and exposed to supplemental 3-h UV-B (+UVB), low R:FR (+FR), or low R:FR and UV-B (+FR +UVB) compared to seedlings maintained under WL as control. Error bars represent SEM of three independent biological replicates. Asterisks indicate a significant difference in transcript

abundance compared to that of WL in each genotype ($^*$ p $<$ 0.05; $^{**}$ p $<$ 0.01).
(PDF)

**S4 Fig. UV-B represses PIF5 binding to promoters of shade marker genes.** (A, B) PIF5-HA chromatin association in Col/Pro$_{PIF5}$:PIF5-3xHA (PIF5-HA). Ten-day-old seedlings were grown in long-day conditions under white light and exposed at ZT3 to 3-h low R:FR with or without supplemental narrowband UVB. ChIP-qPCR was performed for (A) *PIL1* and (B) *HFR1* promoters. The numbers of the analyzed DNA fragments indicate the positions of the 5' base pair of the amplicon relative to the translation start site (referred to as position +1). Fragments designated as Pro$_{PIL1\_-1417}$ and Pro$_{HFR1\_-1689}$ contain a G-box, whereas Pro$_{PIL1\_+1816}$ and Pro$_{HFR1\_-202}$ are devoid of a G-box. ChIP of DNA associated with PIF5-HA is presented as the percentage recovered from the total input DNA (% Input). Data shown are representative of three independent biological replicates. Error bars represent SD of three technical replicates. (PDF)

**S5 Fig. Independent repetitions of ChIP data related to Fig 5B and 5C and Fig 6C.** (A-D) PIF4-HA chromatin association in Col/Pro$_{PIF4}$:PIF4-3xHA (PIF4-HA) and *uvr8-6*/Pro$_{PIF4}$: PIF4-3xHA (*uvr8*/PIF4-HA). Ten-day-old seedlings were grown in long-day conditions under white light and exposed to low R:FR with (+) or without (-) 3-h supplemental narrowband UV-B at ZT3. ChIP-qPCR was performed for (A,B) *PIL1* (repetitions of data shown in Fig 5B) and (C,D) *XTR7* promoters (repetitions of data shown in Fig 5C). (E,F) Chromatin association of PIF4-HA in 10-day-old Col/Pro$_{PIF4}$:PIF4-3xHA (PIF4-HA) and *hfr1-101*/Pro$_{PIF4}$:PIF4-3xHA (*hfr1*/PIF4-HA) seedlings grown in long-day conditions under white light and exposed at ZT3 to 3-h low R:FR with (+) or without (-) supplemental UV-B. ChIP-qPCR was performed for the *PIL1* promoter (repetitions of data shown in Fig 6C). Error bars represent SD of three technical replicates. (PDF)

**S1 Table.** Intersection of genes upregulated by low R:FR and repressed by UVB in wild type (Col, S1A Table), *uvr8-6* (S1B Table) and *hfr1-101* (S1C Table) (lists corresponding to Fig 2A). (XLSX)

**S2 Table. Genes upregulated by low R:FR in Col wild type, *uvr8-6*, and *hfr1-101* (lists corresponding to Fig 2B).** (XLSX)

**S3 Table. Intersection of genes upregulated by low R:FR and repressed by UV-B in wild type and genes upregulated by low R:FR and not repressed by UV-B in *hfr1-101* mutant (lists corresponding to Fig 2C).** (XLSX)

**S4 Table. Oligonucleotide sequences used in this study.** (XLSX)

## Acknowledgments

We thank Emilie Demarsy for helpful comments on the manuscript, Ove Nilsson for kindly providing *bop2*-related seeds, and Patricia Hornitschek and Séverine Lorrain for generating *hfr1-101*/Pro$_{HFR1}$:HFR1-3xHA and *pif5-1*/Pro$_{PIF5}$:PIF5-3xHA lines, respectively. RNA-Seq experiments were performed at the iGE3 genomics platform of the University of Geneva (https://ige3.genomics.unige.ch/). We thank Natacha Civic and Céline Delucinge-Vivier of the iGE3 genomics platform for initial bioinformatics analysis of the RNA-Seq data.

## Author Contributions

**Conceptualization:** Eleni Tavridou, Roman Ulm.

**Data curation:** Emanuel Schmid-Siegert.

**Formal analysis:** Eleni Tavridou, Emanuel Schmid-Siegert.

**Funding acquisition:** Christian Fankhauser, Roman Ulm.

**Investigation:** Eleni Tavridou.

**Methodology:** Eleni Tavridou, Emanuel Schmid-Siegert, Christian Fankhauser.

**Resources:** Christian Fankhauser.

**Supervision:** Roman Ulm.

**Validation:** Eleni Tavridou.

**Writing – original draft:** Eleni Tavridou.

**Writing – review & editing:** Emanuel Schmid-Siegert, Christian Fankhauser, Roman Ulm.

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
