## [Decision Letter · Decision Letter 0]

18 Feb 2020

Dear Dr Ulm,

Thank you very much for submitting your Research Article entitled 'UVR8-mediated inhibition of shade avoidance involves HFR1 stabilization in Arabidopsis' to PLOS Genetics. Your manuscript was fully evaluated at the editorial level and by independent peer reviewers. The reviewers appreciated the attention to an important topic but identified some aspects of the manuscript that should be improved.

As you will see from the detailed comments of the reviewers, they consider the findings important and appreciate the well-written manuscript, adding a connecting element to a plausible model how UV exposure overcomes the shade avoidance response. Both reviewers suggest minor improvements that indeed could make the paper even stronger, but these should be easy to incorporate and address.

We therefore ask you to modify the manuscript according to the review recommendations before we can consider your manuscript for acceptance. Your revisions should address the specific points made by each reviewer.

[LINK]

Yours sincerely,

Ortrun Mittelsten Scheid

Associate Editor

PLOS Genetics

Gregory P. Copenhaver

Editor-in-Chief

PLOS Genetics

Reviewer's Responses to Questions

**Comments to the Authors:**

Reviewer #1: Under FR-rich light, as under a canopy, plants exhibit a shade avoidance response (SAS) leading to enhanced elongation growth. The UV-B receptor UVR8 represses the shade avoidance response under UV-B. This mechanism might lead to a rapid stop in the shade avoidance response when plants grow out of the canopy into the direct sunlight. The molecular mechanisms leading to UV-B-induced suppression of the SAS are poorly understood and currently of great interest to plant biologists.

In this manuscript, the authors demonstrate the important function of the transcription factor HFR1 in UV-B-induced suppression of the SAS. HFR1 is an important repressor of the SAS. The authors show that UV-B enhances the activity of HFR1 by stabilizing the HFR1 protein. They further show that HFR1 is indeed necessary for the UV-B-induced suppression of a subset of SAS marker genes. Indeed, UV-B inhibits binding of PIF4 to the promoter of a subset of shade marker genes and this inhibition is to a large extent dependent on HFR1. Because UV-B induced PIF4 degradation is HFR1-independent, HFR1 activity on PIF4 is independent of the regulation of PIF4 stability.

This manuscript very conclusively uncovers the important role of HFR1 in UV-B signaling under shade conditions. The data therefore very nicely link the activities of UVR8, COP1, HFR1 and PIF4. Their results support the conclusions drawn by the authors. The manuscript is well written.

Major comments:

- the molecular phenotype of the hfr1 mutant in +FR+UV-B is very well analyzed and presented. I am wondering about the visible phenotype of an hfr1 mutant in +FR+UV-B in comparison to +FR? I.e. hypocotyl elongation.

Minor comments:

- please provide a statistical analysis of the ChIP data.

- it appears to me that Fig. 6C very nicely demonstrates that UV-B shows a much weaker inhibition of PIF4-binding to the PIL1 promoter in an hfr1 mutant than in the WT – despite no change in PIF4 protein levels in hfr1 vs. WT in +UV-B (Fig. 6A). In my view, this very nicely demonstrates that UV-B – via HFR1 – inhibits the DNA-binding activity of PIF4 – which is consistent with the known activity of HFR1 which heterodimerizes with PIFs. However, the authors do not really make this point in the results section. Is there a reason for that?

Reviewer #2: The present manuscript by Tavridou et al investigates the molecular details of UVB-mediated inhibition of shade avoidance syndrome (SAS). SAS is induced by low red:far-red (R:FR) light ratios indicating neighbouring plants in close proximity and is characterised by the induction of a well-defined gene set by the PIF4 and PIF5 transcription factors, eventually resulting in accelerated elongation. UVB, perceived by the UVR8 receptor, inhibits this response. The authors showed that UVB promotes the accumulation of the HFR1 transcription factor probably via the UVR8-mediated sequestration of the COP1 E3 ubq ligase, which targets HFR1 for degradation in the dark or in shade conditions. HFR1 forms inactive heterodimers with PIF4 and PIF5 transcription factors that attenuates SAS-related gene expression. They showed that PIL1, another transcription factor implicated in the regulation of SAS acts redundantly with HFR1 in this process. This newly identified molecular mechanism along with the previously described UVB-induced degradation of PIF4 and PIF5 have an important role in the termination of elongation once the plant has overgrown shade.

The experiments are well-planned and executed, providing convincing results. The text is focused, easy to read. The figures are informative and of good quality.

I have just a few minor comments and questions to this manuscript. Answers to the questions might be even incorporated in the Discussion.

1. Abstract, line 33: I would say ‘PIl1-mediated inhibition of PIF4 and PIF5 function’, to indicate that this regulation does not target the expression/level of PIFs.

2. I suggest to quantitate signals on all Western-blots (not only the one in Fig.6). Statistical analysis of these data would provide a robust support for the key statements/findings.

3. Fig.S4 seems to be not finalized yet: „devoid G” should be changed to „devoid G-box” (and explain this in the legends), and indicate HA-derived or mock immunoprecipitation in panel A as well.

4. Fig.2B demonstrates that a significant number of genes are induced by FR in the uvr8 mutant, but not in Col. How could the authors explain this if UVR8 was told to have function in UVB only?

5. Fig.2 deals with shade-induced genes. Did the authors identify shade-repressed genes in their RNA-seq assay? If yes, how was the repression affected by the UVB treatment?

**Have all data underlying the figures and results presented in the manuscript been provided?**

Reviewer #1: Yes

Reviewer #2: Yes

PLOS authors have the option to publish the peer review history of their article (what does this mean?). If published, this will include your full peer review and any attached files.

Reviewer #1: No

Reviewer #2: No

---

## [Decision Letter · Decision Letter 1]

26 Apr 2020

Dear Dr Ulm,

We are pleased to inform you that your manuscript entitled "UVR8-mediated inhibition of shade avoidance involves HFR1 stabilization in Arabidopsis" has been editorially accepted for publication in PLOS Genetics. Congratulations!

Yours sincerely,

Ortrun Mittelsten Scheid

Associate Editor

PLOS Genetics

Gregory P. Copenhaver

Editor-in-Chief

PLOS Genetics

Comments from the reviewers (if applicable):

Reviewer's Responses to Questions

**Comments to the Authors:**

Reviewer #1: The authors addressed my comments satisfactorily.

Reviewer #2: The authors have addressed all of my concerns and I accept all of their responses to my specific questions. In my opinion, neither further experiments nor modifications to the text are required.

**Have all data underlying the figures and results presented in the manuscript been provided?**

Reviewer #1: Yes

Reviewer #2: Yes

PLOS authors have the option to publish the peer review history of their article (what does this mean?). If published, this will include your full peer review and any attached files.

Reviewer #1: No

Reviewer #2: No

**Data Deposition**

http://datadryad.org/submit?journalID=pgenetics&manu=PGENETICS-D-20-00017R1

**Press Queries**

---

## [Editor Report · Acceptance letter]

5 May 2020

PGENETICS-D-20-00017R1 

UVR8-mediated inhibition of shade avoidance involves HFR1 stabilization in Arabidopsis 

Dear Dr Ulm, 

We are pleased to inform you that your manuscript entitled "UVR8-mediated inhibition of shade avoidance involves HFR1 stabilization in Arabidopsis" has been formally accepted for publication in PLOS Genetics! Your manuscript is now with our production department and you will be notified of the publication date in due course.

With kind regards,

Jason Norris

PLOS Genetics

On behalf of:
